# Prescribing systemic steroids for acute respiratory tract infections in United States outpatient settings: A nationwide population-based cohort study

**Kueiyu Joshua Lin**[1,2]*, **Evan Dvorin**[3], **Aaron S. Kesselheim**[1]

**1** Division of Pharmacoepidemiology and Pharmacoeconomics, Department of Medicine, Brigham and Women's Hospital, Harvard Medical School, Boston, Massachusetts, United States of America,
**2** Department of Medicine, Massachusetts General Hospital, Harvard Medical School, Boston, Massachusetts, United States of America, **3** Ochsner Health System, Jefferson Parish, Louisiana, United States of America

* jklin@bwh.harvard.edu

**Data Availability Statement:** The data were obtained from IBM MarketScan to be used under specific data use agreement. Although we can not share the data with the readers according to the

## Abstract

### Background

Evidence and guidelines do not support use of systemic steroids for acute respiratory tract infections (ARTIs), but such practice appears common. We aim to quantify such use and determine its predictors.

### Methods and findings

We conducted a cohort study based on a large United States national commercial claims database, the IBM MarketScan, to identify patients aged 18–64 years with an ARTI diagnosis (acute bronchitis, sinusitis, pharyngitis, otitis media, allergic rhinitis, influenza, pneumonia, and unspecified upper respiratory infections) recorded in ambulatory visits from 2007 to 2016. We excluded those with systemic steroid use in the prior year and an extensive list of steroid-indicated conditions, including asthma, chronic obstructive pulmonary disease, and various autoimmune diseases. We calculated the proportion receiving systemic steroids within 7 days of the ARTI diagnosis and determined its significant predictors. We identified 9,763,710 patients with an eligible ARTI encounter (mean age 39.6, female 56.0%) and found 11.8% were prescribed systemic steroids (46.1% parenteral, 47.3% oral, 6.6% both). All ARTI diagnoses but influenza predicted receiving systemic steroids. There was high geographical variability: the adjusted odds ratio (aOR) of receiving parenteral steroids was 14.48 (95% confidence interval [CI] 14.23–14.72, p < 0.001) comparing southern versus northeastern US. The corresponding aOR was 1.68 (95% CI 1.66–1.69, p < 0.001) for oral steroids. Other positive predictors for prescribing included emergency department (ED) or urgent care settings (versus regular office), otolaryngologist/ED doctors (versus primary care), fewer comorbidities, and older patient age. There was an increasing trend from 2007 to 2016 (aOR 1.93 [95% CI 1.91–1.95] comparing 2016 to 2007, p < 0.001). Our findings are

agreement, readers can request data access to the same data by online application at https://www.ibm.com/account/reg/us-en/signup?formid=MAIL-watsonhealthna.

**Funding:** ASK's work is funded by "Arnold Ventures" (Arnoldventures.org), with additional support from "Harvard-MIT Center for Regulatory Science" (https://hmcrs.org/). The funders had no role in study design, data collection and analysis, decision to publish, or preparation of the manuscript.

**Competing interests:** I have read the journal's policy and the authors of this manuscript have the following competing interests: ASK is an Academic Editor on PLOS Medicine's editorial board.

**Abbreviations:** AIDS, acquired immune deficiency syndrome; aOR, adjusted odds ratio; ARTI, acute respiratory tract infection; CDHP, consumer-driven health plan; CI, confidence interval; COBRA, Consolidated Omnibus Budget Reconciliation Act; ED, emergency department; ENT, otolaryngology; EPO, exclusive provider organization; GERD, gastroesophageal reflux disease; HDHP, high-deductible health plan; HIV, human immunodeficiency virus; HMO, health maintenance organization; NSAID, nonsteroidal anti-inflammatory drug; POS, point of service; PPO, preferred provider organization; RCT, randomized control trial; RECORD, REporting of studies Conducted using Observational Routinely-collected health Data; SD, standard deviation.

based on patients between 18 and 64 years old with commercial medical insurance and may not be generalizable to older or uninsured populations.

## Conclusions

In this study, we found that systemic steroid use in ARTI is common with a great geographical variability. These findings call for an effective education program about this practice, which does not have a clear clinical net benefit.

## Author summary

### Why was this study done?

- Prescribing systemic (oral or injection) steroids for acute respiratory tract infections (ARTIs), a practice lacking clear medical justification, has been identified as common in the US.

- However, prior studies of this issue have not addressed time trend or details of steroid use.

### What did the researchers do and find?

- In this cohort study including 9,763,710 patients with an eligible ARTI encounter, 11.8% were prescribed systemic steroids.

- There was remarkable geographical variability: patients in the southern US were 14-fold more likely to receive steroid injections for ARTI than those in the Northeast.

- The prescribing rate of systemic steroids for ARTI almost doubled from 2007 to 2016.

### What do these findings mean?

- Systemic steroid use in ARTI is common with increasing trend over time and great geographical variability.

- These findings call for an effective medical education program to reduce this practice.

## Introduction

Using systemic corticosteroids in the treatment of acute respiratory tract infections (ARTIs) in the outpatient settings is not recommended by clinical guidelines [1–3]. Data from randomized control trials (RCTs) show that systemic steroids are ineffective in the treatment of lower respiratory tract infections [4]. Similar—albeit more limited—data also show the lack of effectiveness of steroid use in the common cold [5] and otitis media [6]. Studies have shown mixed results on whether systemic steroids lead to faster symptom relief in pharyngitis [7], and possibly also in sinusitis [8].

By contrast, one meta-analysis of RCTs showed even a short course of systemic steroids in sinusitis with polyposis could result in a 3-fold increase in the risk of gastrointestinal disturbances and insomnia [9]. An observational study found that acute adverse events associated with short-term use of systemic steroids, including sepsis, venous thromboembolism, and fracture, can occur as early as the first 30 days of drug exposure [10]. Taken together, available evidence and professional society recommendations do not support prescribing systemic steroids for ARTI in ambulatory settings [1–3].

Despite the lack of clear evidence for this clinical practice, one recent review estimated that 11% of adult outpatients with ARTIs across the US were treated with oral steroids and 23% in the state of Louisiana were treated with injectable steroids [11]. If true, such prescribing trends could be putting tens of thousands of patients at increased risk of adverse events without clear clinical benefits [1, 9, 10]. However, these prescribing rates were based on survey data, which may be subject to recall inaccuracy, particularly regarding details of medication use, such as dose and duration of the prescription [12]. Since these results were drawn from a limited number of years (2012–2013 for oral steroids and 2014 steroid injections), there was no ability to assess time trends. Finally, the data on injectable steroids were limited to one state, leaving open the question of whether there was any regional variation in clinical practice patterns.

To provide a more comprehensive assessment of US prescribing of systemic steroids for ARTI, we used a large claims database across 10 years with nationwide coverage and stratified patients by whether they received steroids associated with ARTI diagnosis via oral, intravenous, and intramuscular routes. We also sought to determine the regional differences and other predictors associated with this practice, since prior analyses have revealed that treating ARTI with steroid injections might be a common practice in the southern US [11].

## Methods

### Study design and data sources

This is a cohort study based on retrospective analysis of a large commercial health insurance database, IBM MarketScan, from January 1, 2007, to December 31, 2016. It contains deidentified records of more than 250 million patients, capturing longitudinal, individual-level administrative claims data from the US, including three components: the Commercial Claims and Encounters Database, the Medicare Supplemental and Coordination of Benefits Database, and the Medicaid Database. Data were drawn from large employers, health plans, and public organizations in the US, providing information on plan enrollment, healthcare utilization and expenditures, demographics, integrated records for inpatient and outpatient events (including diagnosis and procedure codes), and pharmacy dispensings. This database includes individuals with private health insurance coverage but not those uninsured or with public insurance as the primary payor. Electronic outpatient pharmacy dispensing records are considered accurate because pharmacists fill prescriptions with little room for interpretation and are reimbursed by insurers on the basis of detailed, complete, and accurate claims submitted electronically [13]. Pharmacy dispensing information is usually seen as the gold standard of drug exposure information compared to self-reported information [12] or prescribing records in outpatient medical records [14]. The Institutional Review Board of the Brigham and Women's Hospital approved the study protocol and patient privacy precautions. This study is reported as per the REporting of studies Conducted using Observational Routinely-collected health Data (RECORD) guideline (S1 Checklist). All analysis plans and definitions were specified prior to study implementation. The definitions of the study variables were based on literature and validation studies [11, 15–18]. The study protocol is available as S1 Text in the supporting information online.

## Study population

The study population was derived from patients aged 18 or older with an ARTI diagnosis recorded in an ambulatory visit between January 1, 2007, and December 31, 2016, without the same diagnosis recorded in the preceding 180 days. Eligible ARTI diagnoses included acute bronchitis, sinusitis, pharyngitis, otitis media, allergic rhinitis, influenza, pneumonia, and unspecified acute upper respiratory infections. To avoid including injectable or oral steroids prescribed in the context of patients with severe arthritis, we excluded encounters associated with rheumatology or orthopedic services as well as those with diagnoses of noninfectious arthritis or spondylosis on the cohort entry date and the preceding 180 days. We excluded patients if they were in nursing home in the 180 days prior to the cohort entry date (drug exposure data not available for these institutionalized patients). To ensure we have sufficient data to assess baseline comorbidities, patients were required to have continuous insurance enrollment and drug benefit coverage during the 365 days prior to cohort entry date. Patients were excluded if aged 65 or older, owing to their eligibility for the federal Medicare program (IBM MarketScan only has Medicare Supplemental but not fee-for-service Medicare claims). In addition, we excluded patients who were prescribed systemic steroids or with the medical conditions in the 365 days prior to the cohort entry date for which systemic steroids may be appropriate. These conditions include asthma, chronic obstructive pulmonary disease, inflammatory bowel disease, malignant neoplasm, organ transplant, interstitial lung disease, urticaria, rheumatoid arthritis, systemic lupus erythematosus, and systemic vasculitis (see S1 Text for definitions of these conditions).

## Patient characteristics

We extracted and adjusted for the following covariates: age, sex, ARTI indications, geographical region, provider type (nurse practitioner, physician assistants, general medicine physicians [internists or family medicine doctors], medical specialists, or otolaryngology [ENT] doctors versus emergency department [ED] physicians), care location (regular office, urgent care, or walk-in retail clinic versus ED), employment status, insurance plan, related prescription drug use (nonsteroidal anti-inflammatory drugs, proton-pump inhibitors, histamine-2-receptor antagonists, antibiotics, antiplatelets, anticoagulants), and multiple comorbidities, including diabetes mellitus, hypertension, stroke, kidney dysfunction, dementia, obesity, heart failure, ischemic heart disease, atrial fibrillation, venous thromboembolism, urinary tract infections, human immunodeficiency virus (HIV) infection/acquired immune deficiency syndrome (AIDS), fractures, prior falls, gastroesophageal reflux disease (GERD), peptic ulcer disease, major bleeding events, bronchiectasis, connective tissue diseases, and remote history of nonseptic arthritis or spondyloarthropathy and a combined comorbidity score [19] (see S1 Text for definitions of these conditions). The baseline assessment period was the 365 days prior to the cohort entry date.

## Study outcomes

The primary outcome was defined as having a dispensing record or a procedure code indicating use of systemic steroids orally and parenterally (including intravenous, intramuscular, and nonspecific injectable forms) within 7 days of cohort entry date. See S1 Text for details of the study outcome definitions. Follow-up began on the cohort entry date and continued until first of occurrence of outcome event, disenrollment from the insurance or drug coverage plan, death, hospitalization, or nursing home admission (IBM MarketScan only has outpatient dispending records that do not capture medication use in the hospital or skilled nursing facility) or 7 days after cohort entry date.

## Primary and secondary analysis

The study outcome was measured as proportion per 100 patients. The association between patient characteristics and the outcome was assessed by univariate and multivariate logistic regression. Secondary analyses were conducted to test robustness of our findings. First, we repeated all our analyses after restricting to use of systemic steroids within 3 days of cohort entry. Second, we excluded patients with nasal polyps, since in the field of ENT, systemic steroids are commonly prescribed to treat sinusitis [20], particularly sinusitis with polyposis [9]. All analyses were conducted using the Aetion platform and R, version 3.1.2.5 (R Foundation for Statistical Computing), which has been previously validated for use in observational studies [21, 22] and for predicting clinical trial findings [23].

## Results

A total of 41,322,229 patients with ambulatory encounters had one of the eligible ARTI diagnoses, and our study population consisted of 9,763,710 patients (mean age 39.6 years, female 56.0%, Fig 1). With a mean follow-up of 6.35 days (standard deviation [SD] = 1.7 days), 11.8% of patients with an ARTI-related ambulatory visit were prescribed systemic steroids (46.1% parenteral only, 47.3% oral only, 6.6% using both routes; Table 1). Among systemic steroid users with detailed medication information available, 45.5% were prescribed prednisone equivalents of <20 mg, 23.5% 20–39 mg, and 30.9% ≥40 mg. Most (84.7%) were prescribed a steroid prescription of 7 days or fewer, 14.8% 8–29 days, and 0.5% ≥30 days (Table 1).

### Systemic steroid use by ARTI indication

Among patients with an ARTI diagnosis in an outpatient setting, those diagnosed with acute bronchitis was associated with the highest odds of receiving systemic steroid (adjusted odds ratio [aOR] 2.70, 95% confidence interval [CI] 2.68–2.72, $p < 0.001$), followed by acute sinusitis (aOR 2.03, 95% CI 2.02–2.04, $p < 0.001$), pneumonia (aOR 1.80, 95% CI 1.77–1.82, $p < 0.001$), allergic rhinitis (aOR 1.74, 95% CI 1.72–1.75, $p < 0.001$), otitis media (aOR 1.43, 95% CI 1.42–1.45, $p < 0.001$), pharyngitis (aOR 1.42, 95% CI 1.41–1.43, $p < 0.001$), and acute upper respiratory infections (aOR 1.23, 95% CI 1.22–1.24, $p < 0.001$). Among those with an ARTI diagnosis, having a diagnosis of influenza was associated with lower odds of receiving systemic steroids (aOR 0.65, 95% CI 0.64–0.66, $p < 0.001$; Table 2).

### Geographical variability

We found remarkable regional differences. Patients seeking care for ARTI in the South were 3.78 times (95% CI 3.75–3.81, $p < 0.001$) more likely to be prescribed a systemic steroid than those cared in the Northeast. Such prescribing differences were more pronounced for parenteral steroids than for oral steroids. The aOR of receiving parenteral steroids was 14.48 (95% CI 14.23–14.72, $p < 0.001$) comparing the South versus Northeast, and the corresponding aOR was 1.68 (95% CI 1.66–1.69, $p < 0.001$) for oral steroids (Fig 2 and Table 3).

### Provider type and care location

Compared to general medicine physicians, ENT specialists were associated with the highest prescribing rate of systemic steroids for ARTI (aOR 1.48, 95% CI 1.46–1.50, $p < 0.001$), followed by ED physicians (aOR 1.16, 95% CI 1.15–1.17, $p < 0.001$), physician assistants (aOR 1.10, 95% CI 1.08–1.12, $p < 0.001$), nurse practitioners (aOR 1.10, 95% CI 1.08–1.11, $p < 0.001$), and medical specialists, who had a lower rate (aOR 0.68, 95% CI 0.67–0.69, $p < 0.001$; Table 2). Compared to ambulatory care office visits, systemic steroids were more

| 41,322,229 patients with an eligible ARTI ambulatory visit | |
|---|---|

**Excluded**

| | |
|---|---|
| Due to Age >=65 | -1641346 |
| Due to lack of insurance enrollment in the preceding 365 days | -20023604 |
| Due to lack of drug data in the preceding 365 days | -4021297 |
| Due to having the ARTI diagnosis in the preceding 180 days | -574873 |
| Due to nursing home admission in the preceding 180 days | -61442 |
| Due to arthritis/spondyloarthropathy in the preceding 180 days | -1596408 |
| Due to use of systemic steroids in the preceding 365 days | -2545990 |
| Due to asthma recorded in the preceding 365 days | -532605 |
| Due to COPD recorded in the preceding 365 days | -119571 |
| Due to IBD recorded in the preceding 365 days | -52215 |
| Due to malignancy recorded in the preceding 365 days | -181431 |
| Due to organ transplant recorded in the preceding 365 days | -4465 |
| Due to interstitial lung disease/sarcoidosis/psoriasis recorded in the preceding 365 days | -88147 |
| Due to urticaria recorded in the preceding 365 days | -55164 |
| Due to rheumatoid arthritis recorded in the preceding 365 days | -42061 |
| Due to SLE/systemic vasculitis recorded in the preceding 365 days | -17900 |

| 9,763,710 patients in the study cohort | |
|---|---|

ARTI=acute respiratory tract infections, COPD=chronic obstructive pulmonary disease, IBD=inflammatory bowel disease, ILD=interstitial lung disease, RA= rheumatoid arthritis, SLE=systemic lupus erythematosus

**Fig 1. Study attrition chart.**

likely to be prescribed to treat ARTIs in urgent care (aOR 1.27, 95% CI 1.26–1.28, $p < 0.001$), followed by ED (aOR 1.19, 95% CI 1.18–1.21, $p < 0.001$), and they are less likely to be prescribed in walk-in retail clinics (aOR 0.65, 95% CI 0.60–0.70, $p < 0.001$; Table 2).

## Comorbidities

Individual comorbidities negatively associated with odds of systemic steroid prescribing were diabetes, kidney dysfunction, liver disease, dementia, venous thromboembolism, major bleeding events, GERD, bronchiectasis, urinary tract infections, HIV/AIDS, and bronchiectasis. Using a validated combined comorbidity score [19], patients with more comorbidities were associated with a lower odds of receiving systemic steroids. Use of antibiotics, proton-pump inhibitors, antiplatelets, and nonsteroidal anti-inflammatory drugs were positively associated with odds of systemic steroid prescribing (Table 2).

## Time trend

Prescribing of systemic, parenteral, and oral steroids for ARTI all increased from 2007 to 2016 ($p < 0.001$ for all three, Fig 3). The prescribing rate for systemic steroids in 2016 was almost

**Table 1. Steroid use for ARTIs.**

|  | Steroid use within 7 days | Steroid use within 3 days |
|---|---|---|
| **Total number of patients** | 9,763,710 | 9,763,710 |
| **Systemic steroids use, *N* patients** | 1,154,378 | 1,092,626 |
| **Prescribing rate, % (95% confidence interval)** | 11.82 (11.80–11.84) | 11.19 (11.17–11.21) |
| **Route of administration** | | |
| Parenteral routes alone, *N* (%) | 532,142 (46.1) | 524,987 (48.0) |
| Oral route alone, *N* (%) | 545,487 (47.3) | 501,601 (45.9) |
| Oral + parenteral, *N* (%) | 76,749 (6.6) | 66,038 (6.0) |
| **Daily dose in prednisone equivalents, *N* (%)*** | | |
| <20 mg | 280,704 (45.5) | 255,971 (45.6) |
| 20–39 mg | 144,875 (23.5) | 131,962 (23.5) |
| ≥40 mg | 190,739 (30.9) | 173,995 (31.0) |
| **Length of supply dispensed, *N* (%)*** | | |
| ≤7 days | 522,192 (84.7) | 479,467 (85.3) |
| 8–29 days | 91,137 (14.8) | 79,964 (14.2) |
| ≥30 days | 2,989 (0.5) | 2,497 (0.4) |

*Percent among patients with "dose" and "length of supply dispensed" information.

Abbreviation: ARTI, acute respiratory tract infection

double that of 2007 (aOR 1.93, 95% CI 1.91–1.94). The corresponding aOR was 1.33 (95% CI 1.31–1.35) for parenteral steroids and 2.74 (95% CI 2.71–2.78) for oral steroids (Fig 3).

## Sensitivity analysis

When restricting to systemic steroid use within the first 3 days of ARTI diagnosis, the prescribing patterns were similar to that for steroid use in 7 days of ARTI diagnosis (Table 1), and the results of all analyses were similar (see S1 and S2 Tables). After excluding patients with nasal polyps, our estimates for all analyses were not materially changed, and ENT physicians were still associated with an aOR of prescribing systemic steroids of 1.47 (95% CI 1.45–1.48) when compared to the general practitioners.

## Discussion

Using a national sample of privately insured US patients over the last decade, we found 11.8% of patient encounters with ARTI resulted in receiving systemic steroid treatments. Such prescribing has almost doubled from 2007 to 2016, with patients far more likely to receive this care—particularly injectable steroids—in the southern US, even though use of systemic steroid treatments for ARTIs lacks clear scientific justification. Providers in the ED and urgent care, as well as ENT specialists, were more likely to be prescribers. Use of steroids for ARTIs has been increasing over time, with as many as 16.3% of US patients with an ARTI diagnosis aged 18–64 years—or 10.6 million people—receiving such treatment in 2016.

Our estimates of systemic steroid use for ARTI were consistent with a prior study that used national survey data and local administrative data [11]. We further quantified such prescribing by route of administration and found a disproportionally high prescribing rates in the southern states than in other states, especially for the parenteral routes [11]. We did not find any meaningful differences in patient demographics, ARTI indications, care settings, provider types, and patient comorbidities that can explain the remarkable geographical variability in steroid use for ARTI (S3 Table). To put geographical variations of patient characteristics into

**Table 2. Patient characteristics and association with use of systemic corticosteroids.**

| Characteristic | Total population, $N = 9,763,710$, $N$ | Receiving steroids*, $N = 1,154,378$, $N$ (%) | Univariate OR | Multivariate aOR** |
|---|---|---|---|---|
| Age categories | | | | |
| 18 to <25 | 1,752,290 | 184,968 (10.6%) | Ref | Ref |
| 25 to <35 | 1,975,722 | 222,489 (11.3%) | 1.08 (1.07–1.08) | 1.07 (1.06–1.08) |
| 35 to <45 | 2,231,358 | 267,660 (12.0%) | 1.16 (1.15–1.16) | 1.17 (1.16–1.18) |
| 45 to <55 | 2,167,879 | 275,439 (12.7%) | 1.23 (1.23–1.24) | 1.25 (1.24–1.26) |
| ≥55 | 1,636,461 | 203,822 (12.5%) | 1.21 (1.20–1.21) | 1.22 (1.21–1.23) |
| Sex | | | | |
| Male | 4,297,795 | 540,236 (12.6%) | Ref | Ref |
| Female | 5,465,915 | 614,142 (11.2%) | 0.88 (0.88–0.88) | 0.87 (0.87–0.88) |
| ARTI indication | | | | |
| Unspecified upper respiratory infections | 2,354,975 | 223,990 (9.5%) | 0.73 (0.73–0.74) | 1.23 (1.22–1.24) |
| Otitis media | 248,978 | 30,179 (12.1%) | 1.03 (1.02–1.04) | 1.43 (1.42–1.45) |
| Sinusitis | 2,398,502 | 359,649 (15.0%) | 1.46 (1.45–1.46) | 2.03 (2.02–2.04) |
| Pharyngitis | 2,272,142 | 243,711 (10.7%) | 0.87 (0.86–0.87) | 1.42 (1.41–1.43) |
| Allergic rhinitis | 1,498,960 | 196,792 (13.1%) | 1.15 (1.15–1.16) | 1.74 (1.73–1.75) |
| Acute bronchitis | 1,442,379 | 260,681 (18.1%) | 1.83 (1.82–1.84) | 2.70 (2.68–2.72) |
| Pneumonia | 235,440 | 28,307 (12.0%) | 1.02 (1.01–1.03) | 1.80 (1.77–1.82) |
| Influenza | 332,623 | 20,548 (6.2%) | 0.48 (0.47–0.49) | 0.65 (0.64–0.66) |
| Region | | | | |
| Northeast | 1,519,033 | 90,238 (5.9%) | Ref | Ref |
| North Central | 2,231,576 | 177,785 (8.0%) | 1.37 (1.36–1.38) | 1.41 (1.40–1.42) |
| South | 4,120,036 | 768,350 (18.6%) | 3.63 (3.60–3.66) | 3.78 (3.75–3.81) |
| West | 1,740,782 | 103,579 (6.0%) | 1.00 (0.99–1.01) | 1.08 (1.07–1.09) |
| Unknown | 152,283 | 14,426 (9.5%) | 1.66 (1.63–1.69) | 1.69 (1.66–1.73) |
| Provider type | | | | |
| General medicine | 8,232,102 | 945,074 (11.5%) | Ref | Ref |
| Medical specialist | 214,225 | 16,995 (7.9%) | 0.66 (0.65–0.67) | 0.68 (0.67–0.69) |
| ED physician | 559,897 | 82,476 (14.7%) | 1.33 (1.32–1.34) | 1.16 (1.15–1.17) |
| ENT physician | 218,938 | 37,103 (16.9%) | 1.57 (1.56–1.59) | 1.48 (1.46–1.50) |
| Nurse practitioner | 372,791 | 53,333 (14.3%) | 1.29 (1.28–1.30) | 1.10 (1.08–1.11) |
| Physician assistant | 164,030 | 19,308 (11.8%) | 1.03 (1.01–1.04) | 1.10 (1.08–1.12) |
| Care location | | | | |
| Regular office visit | 9,034,253 | 1,048,476 (11.6%) | Ref | Ref |
| Urgent care | 328,244 | 48,388 (14.7%) | 1.32 (1.30–1.33) | 1.27 (1.26–1.28) |
| Walk-in retail clinic | 7,467 | 694 (9.3%) | 0.78 (0.72–0.84) | 0.65 (0.60–0.70) |
| Emergency room | 393,746 | 56,820 (14.4%) | 1.28 (1.27–1.30) | 1.19 (1.18–1.21) |
| DM | 560,839 | 54,566 (9.7%) | 0.79 (0.79–0.80) | 0.68 (0.68–0.69) |
| HTN | 1,440,025 | 188,226 (13.1%) | 1.15 (1.14–1.15) | 1.12 (1.11–1.13) |
| Stroke | 48,340 | 5,651 (11.7%) | 0.99 (0.96–1.02) | 0.97 (0.94–1.00) |
| Kidney dysfunction | 40,425 | 4,064 (10.1%) | 0.83 (0.81–0.86) | 0.93 (0.89–0.96) |
| Liver disease | 129,804 | 14,256 (11.0%) | 0.92 (0.90–0.94) | 0.96 (0.94–0.98) |
| Dementia | 11,307 | 1,155 (10.2%) | 0.85 (0.80–0.90) | 0.85 (0.80–0.91) |
| Obesity | 312,085 | 39,220 (12.6%) | 1.07 (1.06–1.09) | 1.00 (0.99–1.01) |
| Heart failure | 30,396 | 3,259 (10.7%) | 0.90 (0.86–0.93) | 1.00 (0.95–1.04) |
| Ischemic heart disease | 179,551 | 23,113 (12.9%) | 1.10 (1.09–1.12) | 1.02 (1.00–1.04) |
| Atrial fibrillation | 46,100 | 4,987 (10.8%) | 0.90 (0.88–0.93) | 0.98 (0.95–1.01) |
| VTE | 16,022 | 1,587 (9.9%) | 0.82 (0.78–0.86) | 0.93 (0.88–0.99) |

(*Continued*)

**Table 2.** (Continued)

| Characteristic | Total population, N = 9,763,710, N | Receiving steroids*, N = 1,154,378, N (%) | Univariate OR | Multivariate aOR** |
|---|---|---|---|---|
| Urinary tract infections | 571,447 | 63,957 (11.2%) | 0.94 (0.93–0.94) | 0.94 (0.93–0.95) |
| HIV/AIDS | 16,785 | 1,360 (8.1%) | 0.66 (0.62–0.70) | 0.62 (0.59–0.66) |
| Fractures | 118,412 | 13,852 (11.7%) | 0.99 (0.97–1.01) | 1.02 (1.01–1.04) |
| Falls | 29,154 | 3,661 (12.6%) | 1.07 (1.03–1.11) | 1.03 (1.00–1.07) |
| GERD | 353,109 | 44,529 (12.6%) | 1.08 (1.08–1.09) | 0.93 (0.92–0.94) |
| Peptic ulcer disease | 19,689 | 2,446 (12.4%) | 1.06 (1.01–1.10) | 0.99 (0.95–1.04) |
| Major bleeding events | 12,886 | 1,238 (9.6%) | 0.79 (0.75–0.84) | 0.86 (0.81–0.91) |
| Bronchiectasis | 1,382 | 109 (7.9%) | 0.64 (0.53–0.78) | 0.70 (0.57–0.85) |
| Connective tissue diseases | 6,245 | 744 (11.9%) | 1.01 (0.93–1.09) | 0.93 (0.86–1.01) |
| Use of NSAIDs | 1,177,851 | 151,542 (12.9%) | 1.12 (1.11–1.12) | 1.09 (1.08–1.10) |
| Use of PPIs | 672,651 | 89,762 (13.3%) | 1.16 (1.15–1.17) | 1.14 (1.13–1.15) |
| Use of H2RAs | 94,435 | 10,486 (11.1%) | 0.93 (0.91–0.95) | 0.98 (0.96–1.01) |
| Use of antibiotics | 2,930,232 | 360,876 (12.3%) | 1.07 (1.06–1.07) | 1.10 (1.09–1.10) |
| Use of antiplatelets | 90,369 | 12,160 (13.5%) | 1.16 (1.14–1.18) | 1.08 (1.06–1.10) |
| Use of anticoagulants | 52,308 | 5,354 (10.2%) | 0.85 (0.83–0.87) | 0.88 (0.85–0.91) |
| Combined comorbidity score category | | | | |
| <1 | 7,625,943 | 896,204 (11.8%) | Ref | Ref |
| 1–2 | 1,707,347 | 209,643 (12.3%) | 1.05 (1.05–1.06) | 0.91 (0.90–0.92) |
| 2–4 | 380,088 | 43,694 (11.5%) | 0.98 (0.97–0.99) | 0.85 (0.84–0.87) |
| ≥4 | 50,332 | 4,837 (9.6%) | 0.80 (0.78–0.82) | 0.71 (0.69–0.74) |
| Employment status | | | | |
| Active full time | 5,508,470 | 651,569 (11.8%) | Ref | Ref |
| Retiree | 503,273 | 58,215 (11.6%) | 0.97 (0.97–0.98) | 0.98 (0.97–0.99) |
| Active part-time | 104,025 | 10,259 (9.9%) | 0.82 (0.80–0.83) | 0.91 (0.89–0.93) |
| Unknown/other | 3,647,942 | 434,335 (11.9%) | 1.01 (1.00–1.01) | 0.99 (0.99–1.00) |
| Insurance plan type | | | | |
| PPO | 5,960,060 | 748,413 (12.6%) | Ref | Ref |
| Comprehensive | 179,535 | 20,230 (11.3%) | 0.88 (0.87–0.90) | 1.07 (1.05–1.09) |
| EPO | 129,760 | 11,460 (8.8%) | 0.68 (0.66–0.69) | 0.85 (0.83–0.87) |
| HMO | 1,431,532 | 116,018 (8.1%) | 0.61 (0.61–0.62) | 0.72 (0.71–0.72) |
| POS | 803,251 | 110,638 (13.8%) | 1.11 (1.10–1.12) | 1.14 (1.13–1.14) |
| CDHP | 550,856 | 73,079 (13.3%) | 1.07 (1.06–1.07) | 0.94 (0.93–0.94) |
| HDHP | 341,572 | 37,251 (10.9%) | 0.85 (0.84–0.86) | 0.82 (0.81–0.83) |
| Others/missing | 367,144 | 37,289 (10.2%) | 0.79 (0.78, 0.80) | 1.00 (0.99–1.01) |
| Year of cohort entry date | | | | |
| 2007 | 1,368,068 | 143,446 (10.5%) | Ref | Ref |
| 2008 | 1,194,950 | 125,948 (10.5%) | 1.01 (1.00–1.01) | 1.09 (1.08–1.10) |
| 2009 | 1,292,822 | 124,794 (9.7%) | 0.91 (0.90–0.92) | 1.04 (1.03–1.05) |
| 2010 | 991,226 | 111,076 (11.2%) | 1.08 (1.07–1.09) | 1.25 (1.24–1.26) |
| 2011 | 1,040,685 | 110,962 (10.7%) | 1.02 (1.01–1.03) | 1.20 (1.19–1.21) |
| 2012 | 1,104,266 | 144,628 (13.1%) | 1.29 (1.28–1.30) | 1.44 (1.43–1.45) |
| 2013 | 837,644 | 103,709 (12.4%) | 1.21 (1.20–1.22) | 1.48 (1.47–1.50) |
| 2014 | 758,101 | 100,744 (13.3%) | 1.31 (1.30–1.32) | 1.57 (1.56–1.58) |
| 2015 | 576,617 | 89,806 (15.6%) | 1.57 (1.56–1.59) | 1.76 (1.75–1.78) |

(Continued)

**Table 2.** (Continued)

| Characteristic | Total population, $N$ = 9,763,710, $N$ | Receiving steroids*, $N$ = 1,154,378, $N$ (%) | Univariate OR | Multivariate aOR** |
|---|---|---|---|---|
| 2016 | 599,331 | 99,265 (16.6%) | 1.69 (1.68–1.71) | 1.93 (1.91–1.94) |

*Within 7 days of an ARTI.

**Adjusted for all the variables listed in Table 2.

Abbreviations: aOR, adjusted odds ratio; ARTI, acute respiratory tract infections; CDHP, consumer-driven health plan; COBRA, Consolidated Omnibus Budget Reconciliation Act; DM, diabetes mellitus; ED, emergency department; ENT, otolaryngology; EPO, exclusive provider organization; GERD, gastroesophageal reflux disease; H2RA, histamine-2-receptor antagonist; HDHP, high-deductible health plan; HIV/AIDS, human immunodeficiency virus/acquired immune deficiency syndrome; HMO, health maintenance organization; HTN, hypertension; NSAID, nonsteroidal anti-inflammatory drug; POS, point of service; PPI, proton-pump inhibitor; PPO, preferred provider organization; VTE, venous thromboembolism

perspective, the observed risk ratio of 3.78 associated with the southern region could only be fully explained by an unmeasured confounder that was associated with both the southern region and steroid prescribing by a risk ratio of 7.02-fold each [24], above and beyond the adjusted factors. By the same formula, only an unmeasured confounder associated with both the southern region and steroid prescribing by a risk ratio of 28.45-fold each could account for the 14-fold increased odds of receiving parenteral steroids in the South [24]. The aORs comparing the southern region to all other regions were far smaller than these required thresholds (S3 Table). Given lack of convincing evidence to guide such practice, it is not surprising we did not identify any objective factors associated with it. The regional difference therefore are most likely related to local culture, physician preferences, and patient expectations. There were some similarities of the regional differences in the use of systemic steroids versus antibiotics for ARTI, in which the highest prescribing rates of both systemic steroids and antibiotics for ARTI were observed in the southern states [25]. Since both practices are potentially inappropriate, future research is warranted to investigate the possible impact and interactions of the two practices on clinical outcomes by region.

There was very little evidence supporting prescribing systemic steroids for ARTI. As for oral steroids, data from RCTs have shown that treating pharyngitis with systemic steroids may shorten time to resolution of sore throat [7]. For acute sinusitis, meta-analysis of RCTs has deemed systemic steroids to be ineffective as monotherapy, and the small benefit in symptom relief when used as an adjuvant therapy with antibiotics could possibly be explained by attrition bias [8]. An RCT also revealed that systemic steroids are ineffective in the treatment of lower respiratory tract infections [4]. All the prior RCTs investigating systemic steroid use in community-acquired pneumonia recruited hospitalized patients; among them, steroids as adjuvant therapy to proper antibiotics were shown to reduce mortality and morbidity only in patients with severe pneumonia but not for those with nonsevere pneumonia, casting doubt on generalizing the effectiveness to the ambulatory settings [26]. There were very limited RCT data in steroid use in common cold (only intranasal steroids were studied, which was shown to be ineffective [5]) and otitis media (only pediatric population was studied, which was found to be ineffective [6]). With questionable benefits and substantial risks [9, 10], treating ARTI with systemic steroids has not been recommended by clinical guidelines [1].

Systemic steroid prescribing rates were the highest in urgent care or ED. Because the only demonstrated benefits associated with systemic steroid use in ARTI is symptomatic relief, it is possible people with more severe symptoms seek medical attention in urgent care or ED settings, leading to steroid prescriptions. We found that people with more complex comorbidities are less likely to receive systemic steroids for ARTI. It is plausible that providers are less inclined to prescribe steroids to these vulnerable populations who are more susceptible to

## Systemic (oral + parenteral) steroids

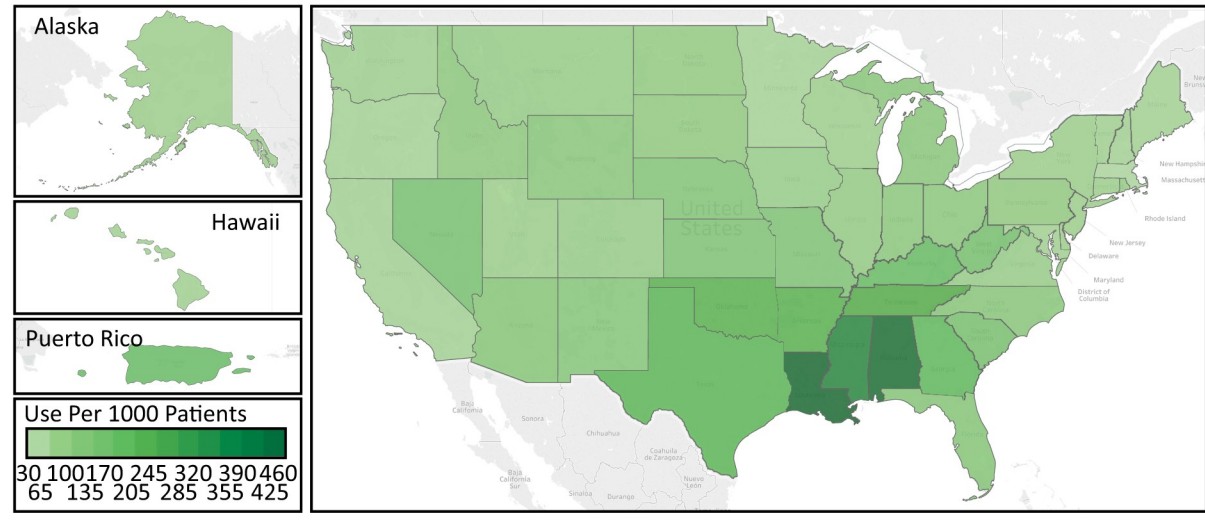

## Parenteral steroids

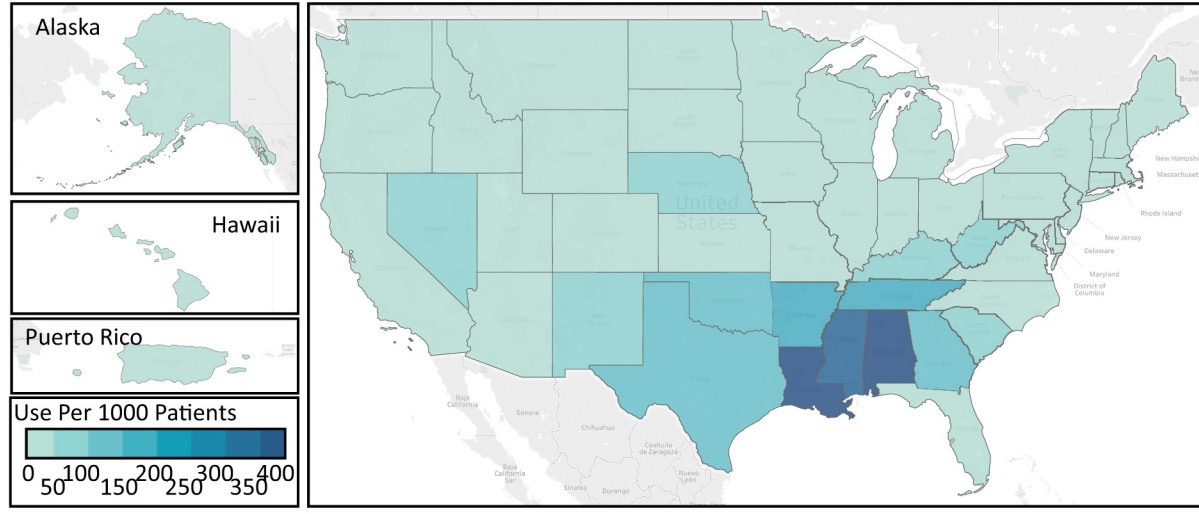

## Oral steroids

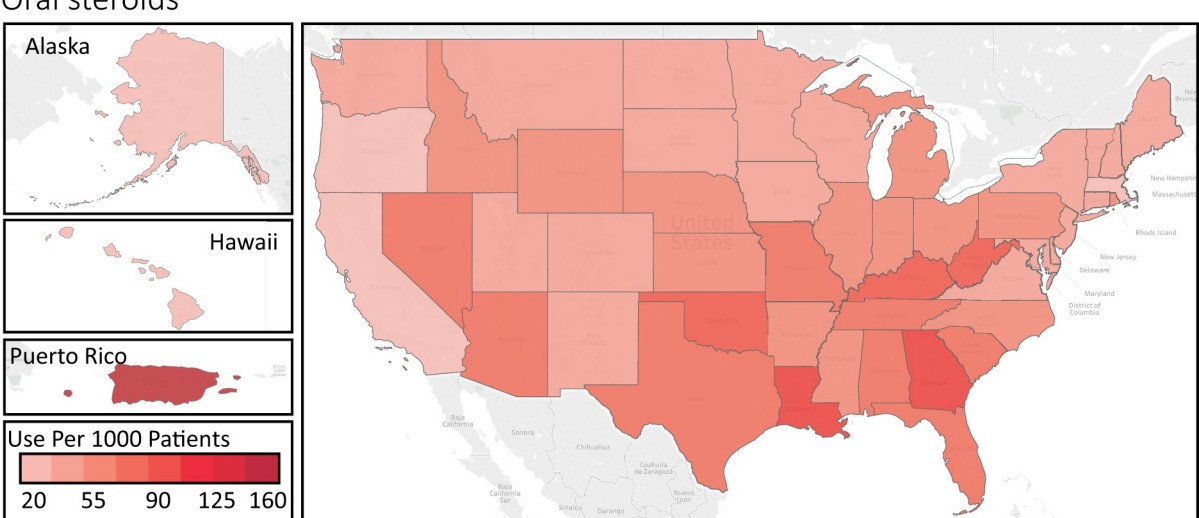

Fig 2. **Geographical variability in prescribing systemic steroids for acute respiratory tract infections.** Source of the base map: https://www.census.gov/geographies/mapping-files/2017/geo/kml-cartographic-boundary-files.html.

developing serious side effects from systemic steroid use [27, 28]. However, adjusting for individual comorbidities, older age was predictive of more steroid prescribing in ARTI. As older age is a strong risk factor for steroid-related complications, including gastrointestinal bleeding [29], sepsis [30], venous thromboembolism [31], and osteoporotic fracture [32], these findings convey an urgent need to reduce this potentially harmful practice.

We also found a steadily increasing trend in prescribing systemic steroids from 2007 to 2016 that was more pronounced for oral than parenteral steroids. The studies showing systemic steroids can lead to faster symptom relief in some limited ARTI indications [7–9] may have encouraged use over time for patients with more severe symptoms. Other factors contributing to this trend could include low prices for steroid prescriptions—for example, numerous oral steroids are often available on $4 generic lists that pharmacies started promoting about a decade ago [33]—and payers' increasingly choosing to integrate patient quality ratings into provider reimbursement. Surveys show that patients often feel better about their physician visits when that visit results in a prescription or other interventions [34], although in this case, the prescriptions do not have supporting evidence behind them.

Our results call for an effective medical education program to help disseminate the messages about the potential risks and limited benefits of steroid prescribing in the context of ARTIs, faithfully reflecting totality of the existing evidence. For example, both physicians and patients should be well informed that treating pharyngitis with systemic steroids may shorten time to resolution of sore throat by about 11 hours [7], at the cost of some potentially serious side effects, including gastrointestinal disturbances, insomnia, sepsis, venous thromboembolism, and fracture, which can occur as early as the first 30 days after a short-term use [9, 10]. Investing in medical education programs to help transform clinical practice in this area would

Table 3. **Associations between geographical region and use of systemic steroids.**

| Variable | aOR (95% CI) |
| --- | --- |
| **Any steroids** | |
| North Central versus Northeast | 1.41 (1.40–1.42) |
| South versus Northeast | 3.78 (3.75–3.81) |
| West versus Northeast | 1.08 (1.07–1.09) |
| Unknown versus Northeast | 1.69 (1.66–1.73) |
| **Parenteral steroids (IV or IM)** | |
| North Central versus Northeast | 2.61 (2.57–2.66) |
| South versus Northeast | 14.48 (14.23–14.72) |
| West versus Northeast | 2.38 (2.33–2.43) |
| Unknown versus Northeast | 3.74 (3.62–3.87) |
| **Oral steroids** | |
| North Central versus Northeast | 1.22 (1.21–1.23) |
| South versus Northeast | 1.68 (1.66–1.69) |
| West versus Northeast | 0.84 (0.83–0.85) |
| Unknown versus Northeast | 1.36 (1.33–1.39) |

Adjusted for all the variables listed in Table 2.

Abbreviations: aOR, adjusted odds ratio; CI, confidence interval; IM, intramuscular; IV, intravenous.

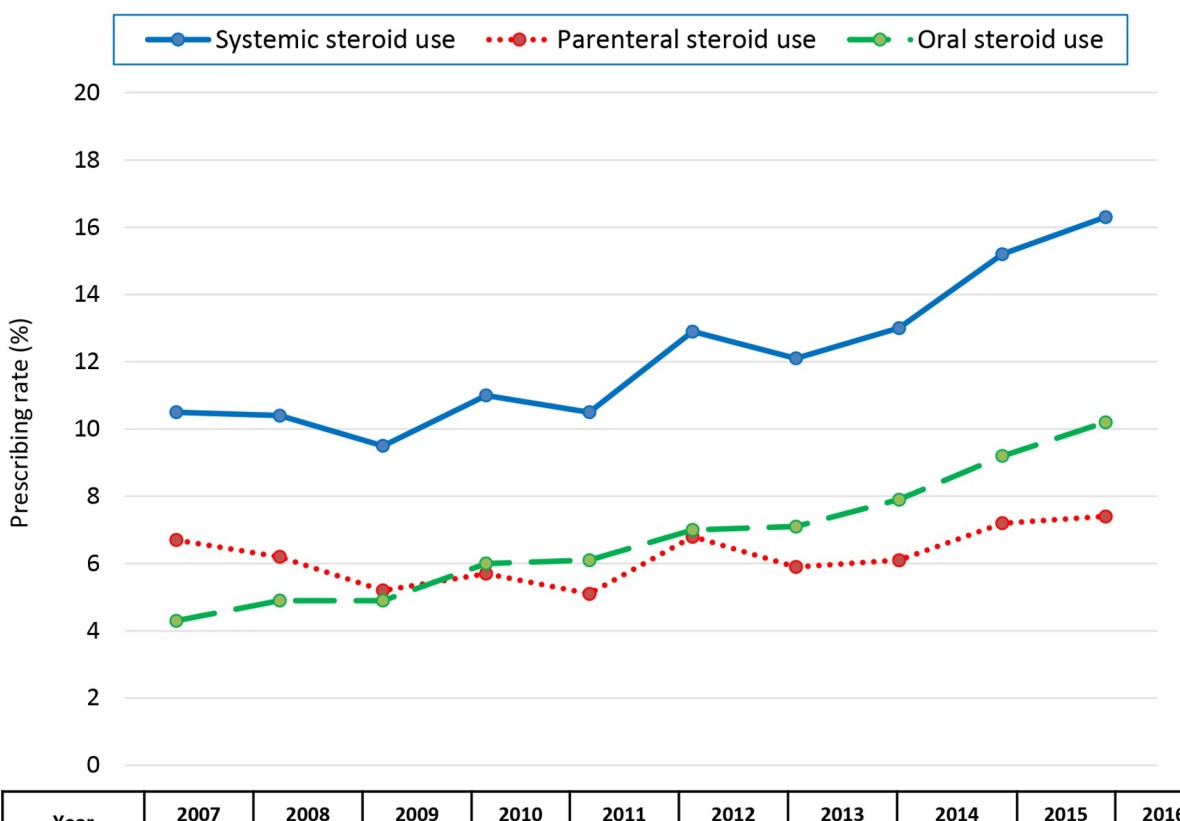

| Year | 2007 | 2008 | 2009 | 2010 | 2011 | 2012 | 2013 | 2014 | 2015 | 2016 |
|------|------|------|------|------|------|------|------|------|------|------|
| Systemic | 10.5 | 10.4 | 9.5 | 11 | 10.5 | 12.9 | 12.1 | 13 | 15.2 | 16.3 |
| Parenteral | 6.7 | 6.2 | 5.2 | 5.7 | 5.1 | 6.8 | 5.9 | 6.1 | 7.2 | 7.4 |
| Oral | 4.3 | 4.9 | 4.9 | 6 | 6.1 | 7 | 7.1 | 7.9 | 9.2 | 10.2 |

**Fig 3. Time trend of prescribing rates of systemic steroids for acute respiratory tract infections.**

improve patient outcomes and reduce health system spending on managing the side effects of such non-evidence-driven care.

Our study has some limitations. First, our primary outcome was systemic steroid prescription within 7 days of ARTI diagnosis, and the indication was not written directly on the prescription or dispensing record. It is possible that some of the prescriptions were not intended to treat ARTI. To minimize erroneous association with the steroid use, we excluded patients with an extensive list of medical conditions for which systemic steroid use may be appropriate as well as those exposed to systemic steroids in the prior year. Also, our sensitivity analysis assessing steroid use within 3 days of ARTI diagnosis showed very similar results (Table 1). Second, we could not stratify our analysis by severity of symptoms because such information was not available in the IBM MarketScan database. However, this limitation should not affect the implication of our findings, as evidence suggests prescribing systemic steroids for ARTI may not be associated with a favorable risk-to-benefit ratio regardless of symptom severity [9, 10]. Third, since we excluded a wide range of steroid-indicated conditions, including asthma, chronic obstructive pulmonary disease, malignancy, and many allergic and autoimmune diseases, our findings cannot be generalized to patients with these conditions as comorbidities. The definitions of the conditions used for inclusion/exclusion criteria were based on prior literature and validation studies [11, 15–18]. As none of these algorithms is perfect,

misclassification of our study variables is possible. However, given that we excluded those who received systemic steroids in the year prior to the cohort entry, our estimated prescribing rates for ARTI is probably conservative. Lastly, our findings are based on patients aged between 18 and 64 years with commercial medical insurance and may not be generalizable to older populations or patients with public health insurance coverage.

Despite these limitations, we found 11.8% of ARTI encounters results in patients being treated with systemic steroids. Such prescribing has been steadily increasing from 2007 to 2016 and is far more common in the southern US. These findings call for an effective medical education program to reduce this practice, which lacks clear scientific justification.

## Supporting information

**S1 Checklist. RECORD Checklist.** RECORD, REporting of studies Conducted using Observational Routinely-collected health Data.
(DOCX)

**S1 Text. Study protocol and definitions of the study variables.**
(DOCX)

**S1 Table. Patient characteristics and association with use of systemic corticosteroids within 3 days of an outpatient diagnosis of acute respiratory tract infections.**
(DOCX)

**S2 Table. Associations between geographical region and use of systemic steroids within 3 days of an acute respiratory tract infection diagnosis.**
(DOCX)

**S3 Table. Patient characteristics by geographical region.**
(DOCX)

## Author Contributions

**Conceptualization:** Kueiyu Joshua Lin, Evan Dvorin, Aaron S. Kesselheim.

**Formal analysis:** Kueiyu Joshua Lin.

**Investigation:** Kueiyu Joshua Lin, Evan Dvorin, Aaron S. Kesselheim.

**Methodology:** Kueiyu Joshua Lin, Aaron S. Kesselheim.

**Resources:** Aaron S. Kesselheim.

**Writing – original draft:** Kueiyu Joshua Lin.

**Writing – review & editing:** Evan Dvorin, Aaron S. Kesselheim.

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
