## [Decision Letter · Decision Letter 0]

6 Nov 2019

Dear Dr. Lin,

Thank you very much for submitting your manuscript "Prescribing of Systemic Steroids for Acute Respiratory Tract Infections in The US Outpatient Settings" (PMEDICINE-D-19-02226) for consideration at PLOS Medicine. 

Your paper was evaluated by a senior editor and discussed among all the editors here. It was also discussed with an academic editor with relevant expertise, and sent to three independent reviewers, including a statistical reviewer. The reviews are appended at the bottom of this email and any accompanying reviewer attachments can be seen via the link below:

[LINK]

In light of these reviews, I am afraid that we will not be able to accept the manuscript for publication in the journal in its current form, but we would like to consider a revised version that addresses the reviewers' and editors' comments. Obviously we cannot make any decision about publication until we have seen the revised manuscript and your response, and we plan to seek re-review by one or more of the reviewers. 

We expect to receive your revised manuscript by Nov 27 2019 11:59PM. Please email us (plosmedicine@plos.org) if you have any questions or concerns.

We look forward to receiving your revised manuscript. 

Sincerely,

Thomas McBride, PhD

Senior Editor 

PLOS Medicine

plosmedicine.org

1- Did your study have a prospective protocol or analysis plan? Please state this (either way) early in the Methods section.

c) In either case, changes in the analysis—including those made in response to peer review comments—should be identified as such in the Methods section of the paper, with rationale.

2- Please ensure that the study is reported according to the RECORD guideline, and include the completed RECORD checklist as Supporting Information. Please add the following statement, or similar, to the Methods: "This study is reported as per the REporting of studies Conducted using Observational Routinely-collected health Data (RECORD) guideline (S1 Checklist)."

The RECORD guideline can be found here: http://www.equator-network.org/reporting-guidelines/record/

3- Please add this statement to the manuscript's Competing Interests: "AK is an Academic Editor on PLOS Medicine's editorial board."

4- Please edit your data statement to provide details on how readers may contact IBM MarketScan for the data used in this study, noting any restrictions that may apply.

5- Please place the study design in the subtitle (ie, after a colon).

6- Please combine the Methods and Findings sections of the Abstract into one section, “Methods and findings”.

7- In the last sentence of the Abstract Methods and Findings section, please describe the main limitation(s) of the study's methodology.

8- Please remove the ® trademark symbol throughout the manuscript.

9- In the Abstract Methods and findings, please include some brief statistics on the included cohort (e.g., age, sex)

10- In the Abstract and going forward, please include p values alongside the 95%CIs.

11- At this stage, we ask that you include a short, non-technical Author Summary of your research to make findings accessible to a wide audience that includes both scientists and non-scientists. The Author Summary should immediately follow the Abstract in your revised manuscript. This text is subject to editorial change and should be distinct from the scientific abstract. Please

 see our author guidelines for more information: https://journals.plos.org/plosmedicine/s/revising-your-manuscript#loc-author-summary

12- In the Introduction, please cite a specific guideline or guidelines re prescribing, rather than "an observational study”.

13- Please include information on ethical approval informed consent in the Methods.

14- Table 2 provides information on gender. Is this accurate or did you mean sex?

15- Though perhaps not the intent, the language of "non evidence based" can be read in rather a sanctimonious way, considering that some of the prescribing appears to be for symptomatic relief in the ED. In the discussion, it could be worth mentioning the alternatives available to doctors for provision of symptomatic relief.

Comments from the reviewers:

Reviewer #1: Alex McConnachie, Statistical Review

The article by Lin et al presents an analysis of routine data from approximately 10 million people, to investigate the use of systemic steroids for acute respiratory tract infections over a 10-year period in the US. This review is primarily focused on the use of statistics in the paper.

The description of the study population is good, giving a clear description of those included and excluded, and why. There is a long list of patient and encounter characteristics that were used to adjust the analyses, all of which appear sensible things to look at. The study outcome is clearly defined. Even though the length of follow-up is not the same for every study participant, given the size of the dataset, and the average length of follow-up, the use of logistic regression is justified. The sensitivity analyses also seem reasonable.

When looking at the associations between ARTI conditions and systemic steroid prescribing, care must be taken, since every patient had at least one condition. So, to say that having influenza was associated with a reduced likelihood of a prescription may not be accurate, since those without a diagnosis of influenza must have had one of the other diagnoses.

Data are presented for parenteral and oral steroids, but the methods section describes extracting the route of administration as oral, intravenous, or intramuscular. Is there a reason for this?

The sensitivity analysis results are reportedly similar to the main analysis, but these results are not shown. Would it be possible to include these results in the supplement, just for completeness?

Given that one of the most striking findings of this analysis is the geographical variation in prescribing rates, I wonder whether it would be of value to carry out stratified analyses, or even to formally test whether the predictors of prescribing vary between geographical regions. For example, the patterns of prescribing in relation to age and gender could be very different in different regions. It may be too much for this paper, but perhaps worth considering.

Table 2 is fine, but I think it could be improved. What matters are the numbers of patients in each row category, and the number and percentage who received a steroid prescription. E.g. for the age group "18 - <25", I would show in the first column, that there were 1,752,290 people in this category, and in the second column, show that 184,968 (10.6%) received a steroid. Since the outcome is receiving a steroid prescription, then as a reader I want to see how the percentage with the outcome varies according to patient and encounter characteristics.

Reviewer #2: This paper uses a large medications database to explore the use of oral and systemic steroids for ARTI in the US. It is a clear and well written paper. I have a number of minor 'style comments' detailed below. For the non US reader a slightly fuller explanation of the claims database might be helpful - what proportion of consultations in the US are captured by this. What is missed in such a dataset (non-insured/other settings?)

The Background is appropriate and justifies the study.

Methods- population at risk clearly described and appropriate for the question. The analysis adjusted for many co-variates although these were not clearly justified in the manuscript, a sentence to justify these would be helpful.

Results

The results are clearly stated including those from the sensitivity analyses.

Discussion.

The main results are summarised clearly with additional information to enable the reader to estimate the population level impact of the steroid prescribing.

The comparison with other literature is appropriate as is the exploration of potential explanations for the variation by setting. 

The final paragraph on p 11 (potential explanations) might be better placed later in the manuscript.

Regarding the first paragraph on page 12 I think it is sufficient to say that the issue should be addressed, and appropriate interventions developed using the models shown to be effective in antibiotic stewardship rather than selecting academic detailing for specific mention.

The limitations section appropriately highlights the reliance on temporal association but replication of findings using the more restricted timeframe is reassuring

Style Comments

Background

I suggest leaving the reader to draw conclusions about whether steroid prescribing is questionable and that the first sentence could simply read

In the outpatient settings is not recommended in clinical guidelines

Para 3- why not say Despite the lack of evidence one recent review….

Some minor wording comments, I dont think 'drastic' is needed as a descriptor and when discussing steroids in sinusitis- the treatment effect could simply be explained rather than 'explained away'

Reviewer #3: This study examines the prevalence and determinants, with a primary focus on geography, of systemic corticosteroid use among over 9.7 million patients with acute respiratory tract infections (ARTIs) meeting inclusion criteria in the United States using large administrative claims databases from 2007-2016. 

1. Novelty: Previous studies have demonstrated high levels of systemic corticosteroid use among patients with ARTI with rates comparable to those demonstrated in this study, albeit using different methods. Could the authors highlight in the discussion where this study truly adds novel findings? 

2. Potential for misclassification: The use of systemic corticosteroids for the treatment of ARTI was the primary outcome. The potential for misclassification may be problematic at the level of the ARTI diagnosis itself and the at the level of systemic corticosteroid indication. The diagnosis of ARTI included acute bronchitis, sinusitis, pharyngitis, otitis media, allergic rhinitis, influenza, pneumonia, and unspecified acute upper respiratory infections. Have the codes for identifying these conditions been validated? If so, please provide references. The assessment indication of systemic corticosteroid involved numerous inclusion / exclusion criteria since systemic corticosteroids are widely used for a variety of indications. Has the approach to assessing systemic corticosteroid use for the purpose of treating ARTIs been validated? If so, please provide a reference. The authors do acknowledge potential misclassification as a limitation. 

3. Generalizability: Approximately 9.7 million patients of over 41 million patients met the inclusion / exclusion criteria (i.e. 24%), which may limit generalizability. Have the authors investigated the generalizability of their findings? They have noted this as a limitation in the discussion section. 

4. A priori definition of outcomes and exposure assessment: Geography was a key exposure that was being assessed yet it is poorly described in the methods section. How was geography defined? It's assumed that this was done at the state level. Did all states have comparable database coverage (i.e. was sampling similar from state to state)? Further, were there any pre-specified hypotheses? Tables 2 and 3 report a very large number of statistical tests. If not, I worry this may simply be a 'fishing expedition' that would be subject to false associations related to multiple hypothesis testing. Inclusion of a formal analysis plan in the methods section is highly advised.

[LINK]

---

## [Decision Letter · Decision Letter 1]

27 Jan 2020

Dear Dr. Lin,

Thank you very much for re-submitting your manuscript "Prescribing Systemic Steroids for Acute Respiratory Tract Infections in US Outpatient Settings" (PMEDICINE-D-19-02226R1) for review by PLOS Medicine.

I have discussed the paper with my colleagues and the academic editor and it was also seen again by xxx reviewers. I am pleased to say that provided the remaining editorial and production issues are dealt with we are planning to accept the paper for publication in the journal.

[LINK]

We look forward to receiving the revised manuscript by Feb 03 2020 11:59PM. 

Sincerely,

Thomas McBride, PhD

Senior Editor 

PLOS Medicine

plosmedicine.org

Requests from Editors:

Please revise your title according to PLOS Medicine's style. Your title must be nondeclarative and not a question. It should begin with main concept if possible. "Effect of" should be used only if causality can be inferred, i.e., for an RCT. Please place the study design ("A randomized controlled trial," "A retrospective study," "A modelling study," etc.) in the subtitle (ie, after a colon).

Please add a study descriptor to the title after a colon, e.g., "... settings: a cohort study" 

In your abstract, please remove the word "remarkable" or substitute "high", say. Please add some summary demographic details and the final sentence of the ‘Methods and Findings’ section should include a sentence on the limitations of the study. 

Please provide p values where 95% Cis are given.

Please begin the "conclusions" subsection of your abstract with "In this study, we found that systemic steroid use ... was common ..." or similar. 

Author summary needs reformatting to bullet points, per house style

Page 4 – please add subheading of Introduction

As the statistician has queried the analysis plan, please add a sentence in the methods to say when it was prepared. 

 p<0.01 to be replaced with p<0.001 or exact values, please

square brackets to be relocated to before punctuation, throughout.

Comments from Reviewers:

Reviewer #1: Alex McConnachie, Statistical Review

The authors have addressed all of my original comments. In Table 2, I was perhaps not clear enough. I was looking for the row percentages, so that, for example, for males, the percentage who received steroids was 12.6% (540236/4297795), and for women, 11.2%. I believe this makes it clearer that men were more likely to receive steroids than women.

In response to one of the editor's points, the authors have provided a protocol. The analysis section of this document is very brief compared to most protocols, and, oddly, is written in the past tense.

[LINK]

---

## [Editor Report · Decision Letter 2]

27 Feb 2020

Dear Dr. Lin, 

On behalf of my colleagues and the academic editor, Dr. Jeremy Goldhaber-Fiebert, I am delighted to inform you that your manuscript entitled "Prescribing Systemic Steroids for Acute Respiratory Tract Infections in US Outpatient Settings: A Nationwide Population-based Cohort Study" (PMEDICINE-D-19-02226R2) has been accepted for publication in PLOS Medicine. 

PRODUCTION PROCESS

PRESS

PROFILE INFORMATION

Thank you again for submitting the manuscript to PLOS Medicine. We look forward to publishing it. 

Best wishes,

Clare Stone, PhD

Senior Editor 

PLOS Medicine

plosmedicine.org